# The role of mHealth intervention to improve maternal and child health: A provider-based qualitative study in Southern Ethiopia

**Girma Gilano** [1]*, **Andre Dekker** [2], **Rianne Fijten** [2]

**1** Department of Public Health Informatics, School of Public Health, College of Medicine and Health Sciences, Arba Minch University, Arba Minch, Ethiopia, **2** Department of Radiation Oncology [Maastro], GROW School for Oncology and Developmental Biology, Maastricht University Medical Centre+, Maastricht, The Netherlands

☯ These authors contributed equally to this work.

\* gilanog@yahoo.com

## Abstract

### Introduction

Maternal and child mortality remained higher in developing regions such as Southern Ethiopia due to poor maternal and child health. Technologies such as mobile applications in health may be an opportunity to reduce maternal and child mortality because they can improve access to information. Therefore, the main aim of this study was to explore the role of mHealth in improving maternal and child health in Southern Ethiopia.

### Methods

This study employed a qualitative study design to explore the role of mHealth in improving maternal and child health among health professionals in Southern Ethiopia from December 2022 to March 2023. We conducted nine in-depth interviews, six key informants' in-depth interviews, and four focused group discussions among health professionals. This is followed by thematic analyses to synthesize the collected evidence.

### Results

The results are based on 226 quotations, 5 major themes, and 24 subthemes. The study participants discussed the possible acceptance of mHealth in terms of its fitness in the existing health system, its support to health professionals, and its importance in improving maternal and child health. The participants ascertained the importance of awareness creation before the implementation of mHealth among women, families, communities, and providers. They reported the importance of mHealth for mothers and health professionals and the effectiveness of mHealth services. The participants stated that the main challenges related to acceptance, awareness, negligence, readiness, and workload. However, they also suggested strategic solutions such as using family support, provider support, mothers' forums, and community forums.

**Data Availability Statement:** All relevant data are within the paper and its Supporting Information files.

**Funding:** Arba Minch University research directorate funded this study with grant number

GOV/AMU/PhDMast/TH3/CMHS/HI/RCO/01/15, awarded to GG. The funders did not play any role in the study design, data collection and analysis, decision to publish, or preparation of the manuscript.

**Competing interests:** The authors have declared that no competing interests exist.

**Abbreviations:** SDG, Sustainable Development Goals; SNNPR, South Nation Nationalities People Region; HAD, Health Development Army; HEWs, Health Extension Workers; II, In-depth Interview; KII, Key Informants' In-depth Interview; FGD, Focused Group Discussion; MCH, Maternal and Child Health.

## Conclusion

The evidence generated during this analysis is important information for program implementations and can inform policy-making. The planned intervention needs to introduce mHealth in Southern Ethiopia. Planners, decision-makers, and researchers can use it in mobile technology-related interventions. For challenges identified, we recommend solution-identified-based interventions and quality studies.

## Introduction

Disparities in maternal and child health are persistent in low and middle-income countries (LMIC). Out of 810 deaths of women per 100,000 live births per day from preventable causes related to pregnancy and childbirth, 94% of deaths occurred in developing and low-income countries. Moreover, out of this 94%, 86% of the deaths occurred in Sub-Sahara and Southern Asia regions [1,2]. For instance, the top five countries with high maternal mortality are Nigeria [67,000], India [35,000], the Democratic Republic of Congo [16,000], Ethiopia [1,000], and Tanzania [11,000] [3]. The United Nations Sustainable Development Goal 3 [SDG-3] endeavors to reduce global maternal mortality to 70 per 100,000 live births, preventable deaths of newborns to 12 per live birth, and children under five years of age to as low as 25 per 1,000 live births by 2030 [4]. If the current mortality is not interfered with through new technological strategies, it might be impossible to achieve SDG-3 in low and middle-income countries [5–7]. The application of mHealth (mobile health) has the potential to remind, inform, and provide remote services to improve maternal and child health [8]. Mobile health is defined as mobile phone-based communications and networking technologies that include mobile phones, patient monitoring devices, personal digital assistants, and other wireless devices used to improve access to health information and support the achievement of health objectives [9]. Mobile health can advance postnatal, antenatal care, child feeding practice, childhood immunization, maternal awareness, healthy child development, and other services in resource-limited countries such as Ethiopia [10].

## Related works

Evidence shows that technology support can improve maternal and child health especially, in highly affected regions such as sub-Saharan Africa, and as there is an adequate number of mothers who can use mobile phones [11,12]. It is a promising advance in LMIC, especially when it has health professionals' support and beliefs [13,14].

Many mothers (93%) improved and maintained child-feeding practices through mHealth support [15]. Mobile health increased hospital delivery from 44% to 80.1% in Kenya [16] and showed improvement in tracking systems that improved maternal and child health in Rwanda [17,18]. The application of mobile phone text and video services improved maternal and child health in Uganda [19], Cape Town [20], South Africa [21,22], Nigeria [23], Pakistan [24], and Ethiopia [25,26].

In Ethiopia, there is a substantial number of mobile phone subscribers (65% nearly 64 million people) which has increased by 1.6% each year since 2021 [27]. Unlike the Ethiopian Health Sector Transformation Plan [HSTP-II] recommendation of mHealth, there is limited information to support the wider implementation [28]. Moreover, the information is more limited in areas of the perspective of health providers, cultural factors, perceived benefits, and

readiness of the health system [29–32]. With these gaps, mHealth implementation may not be possible. Thus, this study aims to explore the health professional's perspective on the perceived role of mHealth on maternal and child improvement, the perceived readiness of health institutions, perceived benefits of mHealth for clients and health institutions, and challenges to implementing mHealth Southern Ethiopian health system. These focus areas may provide information on the possible future successful implementation strategies of mHealth and increase awareness in Ethiopia.

## Materials and methods

### Study setting

The South Nation Nationalities Peoples' Region [SNNPR] is one of the elven regions in Ethiopia. The region is home to more than 55 Ethnic groups living in different administrative and cities. The population of the southern Ethiopian region is 9.126 million [33]. Half of the population is living under the poverty line and it is known for its high fertility rate, and higher infant and maternal mortality next to Somali and Afar [32–35]. Arba Minch Zuria Woreda is one of the districts in SNNPR selected for this study.

### Study design

To explore the role of mHealth in improving maternal and child health among health professionals in Southern Ethiopia, we employed a health provider-based exploratory qualitative study design from December 2022 to March 2023. As part of the large doctoral project on the effect of mHealth on maternal and child health, the role of health professionals is imperative. We focused on socio-demographic, in-depth interviews [II], key informants' in-depth interview [KII], and Focused Group Discussion [FGD] data to describe our objectives. The focus was on maternal health [antenatal care, Institutional delivery, postpartum family planning, and postnatal care] improvements due to mHealth, while child health data focused on information such as child feeding practice and childhood vaccinations [36].

### Participants

**Source population and sample size.** The source population for this study is all health professionals in all primary care facilities in the study area. The targets for the focused group discussion, in-depth interview, and key informants' in-depth interview were physicians, public health officers [higher positions professionals next to general practitioners in Ethiopia's health system], nurses [public, clinical, pediatrics, neonatal, emergency nursing and others], midwifery, and health extension workers [HEWs—community health workers in the Ethiopian context]. The key informants are the facility leaders and HEW coordinators. The primary health care facilities [Health posts, Health centers, and primary hospitals], which are the components of the Primary Health Care Unit [PHCU] with high achievement in maternal and child service, were included. The FGDs were composed of physicians, public health officers, nurses, and midwives for facility workers. Health extension workers or community health workers were included in a separate FGD to avoid the dominance among staff. The relationship between health extension workers and hospital or health center staff is a supervisor-worker relationship, which is why separate data collection is needed.

Four FGDs were conducted with the number of participants ranging from three to eight. In addition, semi-structured interview guides were applied to conduct eight in-depth interviews [II] and six key informant in-depth interviews [KII]. KII is important to comparatively validate the data captured at individual and group levels. Overall, interviews are vital to understanding

various perspectives of realities, and ordinary and extraordinary real-life events that health professionals may not speak about publically. Eight health facilities [two primary hospitals, four health centers, and two health posts] were included to obtain the participants. The study includes every staff who is working in the purposively selected health institutions and those who have experience of at least one year in providing services related to mothers and children. The included work units are expanded programs of immunization [EPI], Mother and Child Health [MCH], delivery, antenatal care, postnatal care, gynecology, outpatient patient department, family planning, health posts, and under-five-year-old care centers.

## Term definitions

**Readiness.** This is the ability of health institutions to provide all necessary resources [human and material] to implement mHealth to improve child and maternal health

**The role of mHealth to improve maternal and child health.** This refers to the use of mobile phones, personal digital assistants [PDAs], patient monitoring devices, and other Information and Communication Technologies [ICT] to support and deliver maternal, and child healthcare services.

**Perceived benefits and challenges.** are positive beliefs of health professionals about the extent, to which mHealth can improve maternal and child health by supporting their service provision and acknowledging the perceived challenges that may threaten its success.

## Data collection and instruments

After a literature review and discussion with respective district health authorities, semi-structured in-depth interviews, key informants' in-depth interviews, and FGD guides were developed. All sections of the guides were in English and then translated to Amharic by local Amharic and English experts. The completed data collection tool contained three sections: the interview, Key informants, and FGD guides in Amharic; since this can easy interviews and discussion process as Amharic is the commonest language to all participants. The principal investigator (GG) collected II and KII. The principal investigator and a trained facilitator conducted FGD. Auto-recording audio devices were used after getting permission from participants to capture the audio data in the Amharic language. Data collectors also took short notes during data collection. The duration of data collection was one and a half hours for FGD and 12 minutes for interviews on average. The data collection guides contain information on benefits, knowledge about mHealth, optimistic views, challenges, cultural barriers, expectations of improvement, concerns, acceptability, privacy and confidentiality, and the future of mHealth. At each session of individual or group data collection, information saturation determined the sample size.

## Data processing and analyses

The sources of data were the II, KII, and FGD. Codes during analysis can come from any of the three methods sources and are summarized, categorized, and coded together based on their similarities to each other (by GG). Depending on the exploratory qualitative nature of this study, thematic analyses are suitable to pursue exhaustive categorization. Two coders evaluated the data coming from the field [Girma Gilano and Tesfaye Feleke]. The authors believe, Tesfaye Feleke's contribution is minimal and cannot grant authorship—coders identified evolving codes from the group and individual level data. Related codes were grouped under similar themes or groups after agreement upon them. Overall, inductive thematic analysis is employed to identify themes, subthemes, and codes by GG. The analysis was performed in Atlas.ti version 23.

### Data quality assurance [Trustworthiness]

The mechanisms such as checking consistency, matching purposes, comparing codes, and refining transcription to control quality were finally approved by people (AD and RF). We tested the data collection tool ahead of the final data collection. The difficult guides to articulate for data collectors and participant understanding were improved. The guide questions, which were difficult to start from, and those concepts that could not be understood by the interviewee were eased for the final data collection (GG). To follow participants' views during transcription and coding, we used the member check method [credibility]. The principal investigator is involved in both coders and transcriptions. We represented the participants' own opinions in their own words while presenting the results to make sure participants' original information was not lost (GG). To ensure dependability and reliability, the authors gave attention to the characteristics of health facilities and participants [37].

### Ethical approval and consent to participate

We obtained ethical clearance from the Institutional Review Board of Arba Minch University and then collected permission letters from the respective Zonal and Woreda/district Health Departments of selected institutions. We secure informed verbal consent from participants. Recording, taking short notes, and taking photographs were all based the permission from participants. Collected information was held confidential and explained to respondents through data collectors accordingly during the collection. Contribution to the assessment was voluntary and all rights of participants were respected. The authors also declare that all steps and activities are performed according to the available international guidelines.

## Results

We conducted four FGDs, nine in-depth interviews, and six key informants' in-depth interviews to see the role of mHealth on maternal and child health according to the health professionals' perspectives. Overall, 35 health professionals from 6 health institutions participated with a mean age of 32.29 years. The professional composition of the study participants is as follows: Nurses [29%, n = 10], Health Officers [20%, n = 7], Midwives [17%, n = 6], and Health extension workers [12%, n = 4]. We stopped data collection after the saturation point of no more new ideas emerging [Table 1].

The analysis is based on 226 quotations collected from different health professionals. We condensed the quotations into 5 major themes and 24 subthemes depending on the emerging relationships. The process was performed in the Atlas ti software. Table 2 below displays the major themes with the corresponding subthemes. We provided all source codes in a supplementary file (Multimedia Appendix 1: Total codes used arranged under their themes). The relationship of each subtheme with the source of data supplementary file 2 (Multimedia 2: Relationship of subthemes with the source of data)

### Acceptability of the mHealth

The health professionals are confident that their colleagues, women, families, and communities will welcome and accept the implementation of mHealth. We categorized health professional's responses into four more subcategories:

**Acceptability: Fitting into the existing health system.** The participants have many reasons to believe that mHealth intervention fits well into the current health system. They think that although the technology is new and may need some time to adapt, they say it is well fit into the current care. They think that there are no cultural obstacles; mHealth is the advanced

**Table 1. The profile of the health professionals who participated in the study in Southern Ethiopia 2023.**

| Methods | Profession | Degree level | Sex | Age | Health institutions |
|---|---|---|---|---|---|
| In-depth Interview | Health informatics | BSc | M | 28 | Shelle Health Center |
| | Nurse | BSc | F | 35 | Shelle Health Center |
| | Midwifery | BSc | F | 42 | Limat Health Center |
| | Midwifery | BSc | F | 26 | Geresse Health Center |
| | HEWs[a] | Diploma | F | 32 | Geresse Health Post |
| | Nurse | BSc | M | 28 | Geresse Health Center |
| | Health Officer | BSc | M | 34 | Geresse Health Center |
| | Nurse | BSc | F | 36 | lante Health Center |
| | Health Officer | BSc | F | 27 | lante Health Center |
| FGD 1[b] | HEWs | Diploma | F | 30 | Sawla Health Post |
| | HEWs | Diploma | F | 28 | Sawla Health Post |
| | HEWs | Diploma | F | 29 | Sawla Health Post |
| FGD 2 | Nurse | BSc | M | 32 | Dilfana Primary Hospital |
| | Midwifery | BSc | F | 36 | Dilfana Primary Hospital |
| | Nurse | BSc | M | 24 | Dilfana Primary Hospital |
| | Nurse | MSc | M | 32 | Dilfana Primary Hospital |
| | Gynecologist | Specialist | F | 30 | Dilfana Primary Hospital |
| | Midwifery | BSc | M | 43 | Dilfana Primary Hospital |
| | Health Officer | BSc | M | 28 | Dilfana Primary Hospital |
| | Health Informatics | Diploma | M | 30 | Dilfana Primary Hospital |
| FGD 3 | Nurse | BSc | F | 31 | lante Health Center |
| | Nurse | BSc | F | 26 | lante Health Center |
| | Midwifery | Diploma | F | 42 | lante Health Center |
| | Midwifery | BSc | F | 29 | lante Health Center |
| | Health Officer | BSc | M | 28 | lante Health Center |
| FGD 4 | Gynecologist | Specialist | M | 31 | Gerese Primary Hospital |
| | General practitioner | MD | M | 40 | Gerese Primary Hospital |
| | Pediatrician | Specialist | F | 34 | Gerese Primary Hospital |
| | Internal medicine | MD | M | 37 | Gerese Primary Hospital |
| Key Informants | Nurse | BSc | F | 39 | Shelle Health Center |
| | Health Officer | MPH | F | 32 | Gerese Health Center |
| | Gynecologists | Specialist | M | 38 | Dilfana Primary Hospital |
| | Health officer | MPH | M | 26 | Lante Health Center |
| | Health Officer | BSc | F | 36 | Limat Health Center |
| | Nurse | BSc | F | 31 | Sawla Health Center |

NB

[a]Health Extension Workers

[b]Focuse Group Discussion.

state of the previous care; existing care culture aspires to technology application; mHealth is the usual health communication through mobile phone; mHealth can improve maternal and child health and community can acknowledge mHealth as an additional effort to improve maternal and child health.

"It fits very well, but I think the process will take some time to adapt" [II, Health Officer (HO), Male (M), age 34].

**Table 2. The major themes and subthemes emerged during data condensation.**

| Major themes | Subthemes |
|---|---|
| Acceptability of the mHealth | • Fitting into the existing health system<br>• Can improve maternal and child health<br>• Can support health professionals<br>• Technology is important to support the system |
| Awareness to use mHealth | • Community Awareness<br>• Family Awareness<br>• Provider Awareness<br>• Women Awareness |
| Benefits of using mHealth | • Reminder or alarming<br>• More effective than the current service<br>• Can help mothers<br>• Can help professionals<br>• Improve decision-making<br>• Improve maternal and child health |
| Challenges to implementing mHealth | • Acceptance related<br>• Awareness related<br>• Device handling related<br>• Negligence related<br>• Readiness related<br>• Workload related |
| Possible solutions for success in using mHealth | • Family members' support<br>• Mothers' forum support<br>• Health professional support |

"Every culture is improving through technology and it is everywhere, so whatever technology has been accepted no opposition" [KII, Nurse, Female (F), age 31].

"I hope they will accept easily because this is part of health communication and no one should oppose because there is nothing new here" [KII, Gynecologist, M, 38].

"Women always want something that allows better service so this can be important" [FGD1.4].

**Acceptability: Can improve maternal and child health.**　Health professionals are willing to accept mHealth because they think that the application of mHealth will improve maternal and child health and support the existing efforts.

*"Improve the existing service in the positive direction" [II, Nurse, F, 36].*

*"We have previous requirements of following patients, so this strengthens the previous idea in practice. This time I don't think anyone will oppose this" [KII, HO, F, 32].*

**Acceptability: Support professionals.**　After clarification on the difference between mHealth and HEW's home-to-home services, participants provided their assessment of mHealth. They express the welcome of mHealth because they believe it may ease their tasks; modernize healthcare provision; strengthen the previous follow-up system; advance health education, be highly evidenced and easy to apply, and easily influence the community. They also underlined that women should come for further investigation as necessary. They also ensured their commitment to bringing awareness to the mothers and the community to achieve mHealth objectives.

*"I think our institution workers will welcome this more than anything. This is modernizing what everyone is eager to have"* [KII, HO, M, 26].

"Health professionals' role should be high to make it accepted by the mothers and community [KII, HO, M, 36].

*"It is interesting to use the technology at this level, so am happy. Mothers usually fear whatever health professionals order and can comply when they order to meet on the phone for the next information"* [FGD 3.2].

*"We have previous requirements of following patients, so this strengthens the previous idea in practice. This time I don't think anyone will oppose this"* [KII, Nurse, F, 39].

**Acceptability: Technology care.**   The participants believe that technology is now part of everyone's daily life. They thought that giving services without technology would decline soon and they are happy to hear about mHealth technology. They also emphasized that automated service is easy to apply and provides support for clients. They expressed a good attitude to have the technology to solve some client problems.

*"Previous services are not different but this is automated, evidenced, and easy to apply"* [II, Health Informatics (HI), M, 28)

*"Am happy that finally, we will have the technology to support our client, especially for appointments"* [KII, Nurse, F, 39].

"People are currently eager to have new technology so if no payment for the technology not everyone will be happy to have these mothers. It can also reduce excessive human power" [KII, HO, F, 32].

"Accept the technology as part of easing their work concerns. Awareness creation among mothers and community is critical" [FGD 7.1].

## Awareness themes

**Awareness creation: Community, family, and health care provider awareness.**   Health professionals believe that awareness is a key factor for any new technology to succeed. The reason is that community awareness can improve all works within the community and is an opportunity to create awareness with the public even for other health goals such as providing care, treatment, and consultation. In individual families, older children and husbands' awareness was mentioned as a key strategy to delivering the service. Additionally, they provided their feelings regarding the effect of healthcare provider awareness on the mother's overall service uptake. They said healthcare provider awareness is key to influencing women and the community's attitudes, improving health institutions' performance, and motivating health professionals. They emphasized the importance of using the Health Development Army [HDA] and mothers' forum and community mobilization as the strategies. HDA is a political structure that creates health-concerned groups. These groups bring all related health problems in the community to the government and coordinate, and motivate health-related community discussions in each village in Ethiopia. Mothers' Forum is a monthly discussion of mothers with government officials and health authorities on the matter of maternal and child health, and sanitation.

"If one mother served all will be interested to take and that will also motivate households to prepare mobile for every mother who gets pregnant" [II, HEW, F, 32].

"Health development community members are the chance to convince mothers and monitor. Currently, due to the sanction, we are suffering a lot, but when everything becomes all right, I think this the best thing our community deserves." [KII, HO, M, 36].

*"Using mHealth consultation, providing care, or treating patient or client look special because we can do whatever when we can or at any time"* [FGD 4.2].

"In health institution awareness training for the health profession, increasing positive feedback mechanism"[KII, HO, M, 26].

**Awareness creation: Women's awareness.** Initial awareness of using mHealth was iterated among health professionals, to sum up the importance of women's awareness. Participants are confident that mothers will use mHealth respectfully when things get right at the initial contact. However, some participants argued that support should be continuous as long as the service exists. When the initiation is successful, it can be accompanied by behavioral change and the community itself will promote the service for pregnant mothers.

*"After mothers understand the service, they will use and since they have access, it will improve the service. First mothers need awareness after that the service can be effective" [*II, Nurse, F, 35].

*"Maybe this mobile service can be accompanied by behavioral change education"* [FGD].

"Throughout mothers may need support. We can help with understanding. If the mother can read, has a mobile, able to understand there will be no more obstacles" [II, Nurse, F, 36].

*"The way of following success and promotion should be continuous. Mothers can be encouraged even to use and buy a mobile phone to be pregnant and this can be common practice for women in the community. They may even ask a woman why she was pregnant before having a phone"* [FGD 2.4].

*"We have to teach mothers how to use mobile, what is mHealth, how to open, and others. If all input resources are fulfilled, there will be no problem"* [II, Midwife (MW), F, 26].

## Benefits of the mHealth Theme

**Benefits of mHealth: Alarming.** During data collection, participants revealed that every system is being digitized and digitalization of health is vital to automate client services. Consequently, they reported the significance of using mHealth for rural mothers to provide information for decision-making. This enables mothers to get a reminder, information on specific services, and signal health danger signs.

"I think mHealth can improve access, appointment on time, availability of information for decision making increased"[II, HEW, F, 32].

"Every system is getting digitized so digitizing health may have much advantage more than just appointment"[II, MW, F, 42].

**Benefits of mHealth: Effective.** According to the perceived effectiveness of mHealth, participants reported that mHealth might reduce resource wastage, can change community service-seeking behavior, and can advance existing services.

*"If all input resources are fulfilled, there will be no problem. It is not a chemical and is not harmful and it is helpful. I don't see any obstacles because we are given them by aware mothers, preparing how mothers can read and all other necessary things to fulfill the service needs"* [II, Nurse, M, 28].

*"It will be very effective because it advances the services"* [II, HO, M, 34].

**Benefits of mHealth: Help mothers.**   Participants strictly believe benefits can only be understood after the advantages and drawbacks of the intervention are clearly defined for mothers. Health information accessibility, family knowledge of the need for service from messages, change of community health, reduced transportation for mothers, filling the human gap, change of community perceptions, and healthy generation are the reasons participants believe using mHealth is more beneficial than current care.

*"Over time this can change community perception and they may reebok mothers who faced the health problem because of poor follow up and poor usage of messages because they already knew that they are under intensive attention"* [FGD 2.4].

*"Support related to mHealth, fulfillment of deficiencies of human and other resource limitations, fulfilling the maternal and child health vision, and respect from the community, and acceptance from the community"* [8.4].

"It is a big opportunity to meet the patient again virtually, especially when you missed something in person. The consultation normally should not be a one-time activity rather an effective consultation should be continuous."[FGD 4.3].

"The difference with mHealth is that mothers can always get counseling or health education unlike that of visits when a large queue limits their counseling and professionals are not stable. I think mHealth mothers can be benefited if the current behavior improves"[FGD1.3].

**Benefits of mHealth: Help professionals.**   As HEWs cannot visit every household every time, participants believe that mHealth might help mothers get better service, give further service improvement, receive services at home, and child feeding information, revive maternal and child care, and technology experience for all services.

"We always give our best service to our best capacity and mHealth may advance the service and ease our load "[II, HI, M, 28].

"It is helpful to also send information which is not appropriate in person "[KII, HO, F, 32].

**Benefits of mHealth: Improve decisions.**   The use of mHealth increases access to health information to make decisions. It reminds people when to take service and allows clients to make their own decisions on their matters. Participants in this study argued that the availability of triggering information and the alarming system could contribute to improved maternal decision-making. However, they need some awareness of its benefits, risks of not using the available information, alerting for danger signs, and satisfaction.

*"It will improve mothers' time wastage for information that they can get through mHealth"* [II, MW, F, 26].

"It can improve their decision-making. This service could increase women's self-decision especially, on ANC, PNC, and feeding practice" [II, Nurse, M, 28].

**Benefits of mHealth: Improve mother and child health.**   Participants reported that mHealth could even improve previous service negligence, and increase service utilization, and interest of the mothers. If it is applied continuously, it can improve the current service one step forward.

*"Non-use of the service because of negligence and forgetting can be improved by frequent messages"* [II, HO, M, 34]. *"It can increase utilization better than that we have previously"* [II, MW, F, 42].

*"If it continues, it can be more effective and be more important than the previous way of service provision"* [II, Nurse, M, 28]. *"So, in that case, it can completely change the stream and interest. Or they may consider the attention they are getting"*[FGD2.3].

## Challenges of mHealth implementation themes

**Challenges of mHealth implementation: Acceptance.** Ensuring the importance of mHealth to improve maternal and child health, participants name some challenges. Among the reiterated challenges, information security issues can occur when the mother cannot read the message herself. Trying to use mHealth without the husband's awareness, recognition of pregnancy only after the third month in some areas, sustainability of the mHealth intervention, installation of computers, and lack of support shortly after implementation are mentioned as other challenges to accept mHealth.

"I think it will improve service but if someone gets information from her phone, she should be someone who is actively using mobile"[II, HI, M, 28]. "Sometimes people may think mobile phones make women rude" [II, Nurse, F, 35]. "Sometimes our community may not say pregnancy until the conception reaches third months, so the application of mHealth may declare it early which may cause some inconvenience" [II, Nurse, M, 28].

"Reading is one of the difficulties. Maybe the absence of electricity, inability to read, lack of mobile phone, lack of interest, lack of support, uninviting political environment" [II, HEW, F, 42].

**Challenges of mHealth implementation: Awareness.** Participants pointed out that some challenges such as prelacteal feeding and zero-level milk being given at hospitals when the mother is not awake need a behavioral change in addition to other challenges such as maternal education, cell phone usage, and poor economy. They also suggested counseling, involving the husband in maternal and child health, and more community awareness.

*"There is always an obstacle but can be reduced by awareness creation. Readability, education, mobile access, network, and refusal may be problems"[II, MW, F, 26]."We know usually even hospitals allow zero-level milk and that may be confusing for a mother to give milk at eerily time"[II, HI, M, 28]. "Here when a child is born it feels like good responsibility to give diet immediately"* [II, Nurse, F, 35].

**Challenges of mHealth implementation: Devices handling.** Despite the lack of phones and loss of phones during follow-up, there is a challenge of manipulating phones to read messages and the sustainability of the program. They express a certainty that these and other challenges are solvable if not for the time constraints.

*"Sustainability of the program is the concern, who will always look after the computer" [FGD 10.1]. "All of them are solvable but may need time to solve" [II, MW, F, 26]. "But not all mothers have mobiles. Sometimes they may lose their mobile while on appointment and there is still the chance of losing mother even with mHealth" [II, Nurse, 36].*

**Challenges of mHealth implementation: Negligence.** In rare cases, participants reported that few mothers could be negligent, reluctant, and careless. However, they also provided possible solutions that at the initiation of any program some challenges are expected could improve over time through awareness creation for behavioral change.

"But some are negligent, and mothers knew the consequence of not complying" [II, HEW, F, 32].

*"I don't think there will be a problem unless the carelessness of professionals affected those mothers' acceptance; they might say is there anything like that?" [II, MW, F, 42]."Reluctance, being bored of many messages over time, preoccupation with important things, hand-to-mouth lifestyle, and daily laborer work" [FGD9.3].*

**Challenges of mHealth implementation: Readiness.** Infrastructure and newness of the service were reported to be a readiness challenge. However, they emphasized the ownership and continuity of service.

*"I have to fulfill resources to start this service. Awareness to health professionals is necessary"* [KII, HO, M, 26].

*"But we have to stress continuity and ownership; otherwise when there is an interruption mother may lose hope and distribute the information"* [II, Nurse, M, 28].

**Challenges of mHealth implementation: Security.** Participants expressed information security concerns when someone else reads a message meant for the mother; however, they also reported that sharing between the mother and health professional and even family is of no concern.

*"Confidentiality and privacy should be kept by a person who is registering information i.e. understanding of the message should be considered otherwise mother may go for who can read the message" [II, HEW, F, 32]."If everyone can access the information every woman accepts, there could be a little security issue. The information going to Mother's hand not anywhere else so why we do fear" [II, Nurse, F, 35]*

**Challenges of mHealth implementation: Workload.** Participants argued that out of all the possible challenges that mothers might have, workload took the larger share compared to household members. However, they also provided the insight that the workload cannot preclude taking mHealth.

*"By the way, mothers know their appointment but it is just the load of work they have every day. Unless she says I am sick, she cannot be allowed to go to the health institutions. It looks like there is no way to change but there should be some strategies for mothers of the daily laborer"* [II, HI, M, 28].

## Solutions theme

**Solutions: Family help.** Despite reading challenges, there are learned children in every household and the women can read the local language in most cases. Thus, participants iterate by increasing household awareness, promoting community-level women's education,

involving husbands, and improving maternal education through Golmasa Timhert [adult education]. Adult education centers are available in almost every village in Southern Ethiopia and the attendants have no age limitations.

"Most women read the local language so if it can be in the local language it will be helpful. This means more than half of mothers can use this service. Almost every husband has a mobile phone and there are learned children in every house to solve the maternal inability to read. A mobilizing community for maternal education and behavioral change is important" [II, HEW, F, 32].

**Solutions: Forum help.** Participants suggested ways to promote women's education through events such as women's forums, community forums, community wing, HDA, 1 to 5 political structure [6 people group], and Golmsa Timhert [village adult education centers]. The community wing has quarterly discussions and reports with community members regarding health service provision.

"Using mothers forum, HDA, voluntary health team, community wing movement, one-to-five [1 ∧ 5], and using HEWs usual program. Mothers' forums will be a nice thing to increase awareness of mothers. Using HEWs can also help the mother become aware of mHealth care. In villages without electricity, there are solar centers for charging mobiles so no problem; just we can support those centers" [II, MW, F, 42].

"Using a voluntary health team in each community to promote mHealth implications; using every contact to promote the mHealth service; working in collaboration with other stakeholders for resources and support and using politics/government in areas of difficulty to support community mobilization." [FGD11.4].

"Maternal education can be improved through adult education [Golmasa Timhert]. Educating females as a community and using women's forums to reach and aware mothers" [FGD 11.5]

**Solutions: Professional support.** Health professionals indicate enthusiasm for a possible individual and community-level behavioral change due to this program, maybe also for services other than maternal, and childcare. The behavioral change might be important specifically for mothers with multiple previous pregnancies.

*"It can change the community behavior and they may then become dependent on mHealth" [II, MW, F, 26]."Except for primigravida, all para gravida mothers knew the consequence of not using the counseling information in enough amounts. It is important to maintain that information flowing but behavioral change could be very important to solve this problem" [FGD2.3]. "Using HEWs to reach lost or bored mothers. Considering children for that mother is not learned. Additionally, the use of spouse phone is a big alternative, Involving partner in maternal and child care, and use of children phone" [FGD11.2].*

## Discussion

Application of mHealth is an opportunity in low and middle-income countries such as Ethiopia. It can improve maternal and child health with a relatively low cost of service [38]. This study evaluated health professionals' perspectives on mHealth implementation to improve maternal and child health. It provides insight into the barriers and facilitators perceived by

health professionals when introducing mHealth to improve maternal and child health in Southern Ethiopia. It summarizes health professionals' perspectives regarding the application of mHealth for maternal and child health, its effectiveness, challenges, awareness, and opportunities. This study also discussed the readiness of the current health system to accommodate mHealth. Overall, we found an optimistic view from participants regarding the possible role and acceptance of mHealth to health professionals to improve maternal and child health. The health professionals also acknowledged the important challenges to may be encountered implementation of mHealth technology.

Health professionals reported that mHealth is seen as an advancement in existing healthcare, not a paradigm shift. They argued that more awareness is needed to commence the program; however, it has similar objectives and might not require radical behavioral change. They believe there are no cultural barriers that can obstruct its implementation, so it fits into the current healthcare provision system of Ethiopia and can be an important support for the existing health service. This is also similar to the opinions in other studies conducted on mHealth introduction into the existing healthcare services [39,40], which may indicate the need for minimal measures to implement mHealth and may reduce the time for familiarization and acceptance. The participants suggested that, even with simple messages such as reminders, it is possible to increase maternal responsibility which can improve care utilization and service uptaking.

Health professionals also reported that mHealth could ease their tasks and modernize services connection to clients. They believe there could be a substantial improvement in the uptake of maternal and child healthcare and the workload of healthcare providers such as home visits, campaigns, providing health education, and easy follow-up. They believe mHealth can reduce the time for supportive supervision, reporting, capturing data, and monitoring time especially, for community health workers [40]. This shows that healthcare providers are willing to accept the introduction of mHealth into the current healthcare system to improve maternal and child health. The participants emphasized the importance of awareness creation for the community, families, mothers, and health professionals themselves for the success of mHealth. They also provided strategies such as using the Health Development Armies [HDA], mothers' forums, community voluntary health teams, community mobilization, community wings forums, and 1 to 5 networks, which are common to carry out political and community activities in the current political system. The 1 to 5 is a six-person group throughout the community, initially set for political purposes but now used for health information dissemination. Creating maternal awareness on mHealth might motivate health service utilization, increase families' support for mothers to utilize health services, and increase community awareness to support all mothers to benefit from health services. Similar information was found in the previous literature [39,41–43], which may indicate that awareness creation to implement mHealth is a significant strategy.

Participants have reasons to believe mHealth might facilitate the improvement of maternal and child health. They think that mothers might miss fewer appointments and can be alerted to health danger signs through messages. The participants reported that mHealth could be resource-effective and can change community health-seeking behavior. It can save time, reduce travel costs, fill in-person limited human gaps, change community perceptions, increase accessibility, improve maternal decision-making, and create a healthy community. This information is consistent with other qualitative opinions [40,44] and the mHealth strategy to improve maternal and child health improvement in Ethiopia.

Although participants discussed many benefits and strategies of mHealth interventions, they also named possible challenges such as less integrated family health, resource limitations, poor awareness, and readiness of the health system. For these challenges, they underlined the

need for key solutions such as involving husbands, families, and communities in mHealth-based service provision. If the husband and family members are not aware that the wife is receiving some messages, they may become suspicious. The additional challenge is women's lack of education to read messages. The women may use other people to read messages; however, this might reduce the women's comfort due to the lack of privacy. The woman's capacity to adapt to technology and her ability to manipulate mobile phones might add further challenges. Finally, participants questioned the current readiness of health institutions to implement mHealth, resource limitations, and continuity after implementation started. Many previous studies acknowledge the presence of these challenges but also put substantial evidence of mHealth success in the face of these challenges [39,40,43,45–47]. This encourages researchers to continue working with mHealth to improve maternal and child health challenges in our contexts.

The participants also provided strategic solutions for the reported challenges. They suggested that awareness among husbands, family members, and the community might reduce the challenges. The use of the local language and Golmasa Timhert [adult education] could enable mothers to read messages. Furthermore, the participants suggested unique strategies such as using HDA, voluntary health teams, community forums, community wing forums, HEWs, 1 to 5 networks, community-wide mobilization, and health professional support to improve the challenges. Other solutions such as training, technical support, economic support, creating an easy-to-use system at both ends and contextual management, and defining implementation strategies. The available literature shows consistency with our participant's opinions [40,48,49]. Participants are confident that the solutions they suggested are the possible remedy for the challenges and encourage implementation.

## Strength and limitation

A limitation of this study is that it was conducted in Southern Ethiopia with a limited number of participants. As such, it only shows the view of the study participants in their local context. Despite the limitations, this study provided substantial evidence of challenges and their remedies to implement mHealth to improve maternal and child health. It can be transferred [applied] to a similar context.

## Conclusion

According to the evidence generated during in-depth interviews, key informant in-depth interviews, and focused group discussions, the findings about mHealth can be the base for mHealth program implementations and can inform policy-making in the field. The participants revealed the need for effective mHealth applications in underdeveloped African countries such as Ethiopia to improve maternal and child health. The benefits of this application include reminders, automated health education, less resource consumption, and information on health danger signs. They also provided information on some challenges such as the readiness of the health system to accommodate mHealth, maternal education, sustainability of the program, and continued support for the users. During the data collection discussions and interviews, the participants ascertained the possible remedy for the challenges they portrayed. They underlined the need for the use of forums that engage women's, community, HDA, and community voluntary health teams. Political structures such as 1 to 5 and community wing movements could also enhance implementation since they create awareness.

In addition to the importance of the synthesized evidence for the planned intervention, the authors believe that the evidence is vital to planning, decision-making, and policy decisions based on this evidence. It shows the acceptability of mHealth which is the base for policy

decisions. Depending on the nature of the technology and intervention, we also recommend additional studies to generate more evidence for planning and decision-makers.

## Future implications

Despite the local and specific contextual nature of this study, our findings highlighted the role of mHealth in improving maternal and child health, especially in rural areas. Rural areas such as Southern Ethiopia have many limitations in the context of access to services, transportation access, and disparities in healthcare distribution. Many areas are hard to reach where mothers cannot access basic maternal and child healthcare. However, health professionals reported that it is possible to provide a full extent of service on the phone and alert mothers to be aware and come for care through messages. Since health professionals are at the heart of the mHealth implementation, their positive response shows the potential success of the future implementation. Since the previous studies focused on women's willingness and service uptaking in response to mHealth, the current study filled the remaining gap and paved the way for wider implementations.

## Supporting information

**S1 Checklist. STROBE statement—checklist of items that should be included in reports of observational studies.**
(DOCX)

**S1 Data.**
(DOCX)

**S1 File.**
(ZIP)

## Acknowledgments

The authors are thankful to Arba Minch University for its logistic support for the study and grateful to qualitative expert Mr.Tefaye Feleke [Assistant professor] and Sewunet Sako for support with software management and coding.

## Author Contributions

**Conceptualization:** Girma Gilano, Andre Dekker, Rianne Fijten.

**Data curation:** Girma Gilano, Rianne Fijten.

**Formal analysis:** Girma Gilano.

**Investigation:** Girma Gilano.

**Methodology:** Girma Gilano, Andre Dekker, Rianne Fijten.

**Project administration:** Girma Gilano.

**Software:** Girma Gilano, Andre Dekker, Rianne Fijten.

**Supervision:** Girma Gilano, Rianne Fijten.

**Validation:** Rianne Fijten.

**Visualization:** Girma Gilano.

**Writing – original draft:** Girma Gilano, Andre Dekker, Rianne Fijten.

Writing – review & editing: Girma Gilano, Andre Dekker, Rianne Fijten.

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
