## [Decision Letter · Decision Letter 0]

29 Sep 2023

PONE-D-23-11693What do health professionals think about the role of mHealth in improving maternal and child health: A provider-based qualitative study in Southern EthiopiaPLOS ONE

Dear Dr. Gilano,

Thank you for submitting your manuscript to PLOS ONE. After careful consideration, we feel that it has merit but does not fully meet PLOS ONE’s publication criteria as it currently stands. Therefore, we invite you to submit a revised version of the manuscript that addresses the points raised during the review process.

We look forward to receiving your revised manuscript.

Kind regards,

Daniel Dramani Kipo-Sunyehzi, PhD, MPhil, MA, BA

Academic Editor

PLOS ONE

Journal Requirements:

4. Please ensure that you refer to Figure 1 in your text as, if accepted, production will need this reference to link the reader to the figure.

5. We note that Figure 1 in your submission contain map images which may be copyrighted. All PLOS content is published under the Creative Commons Attribution License (CC BY 4.0), which means that the manuscript, images, and Supporting Information files will be freely available online, and any third party is permitted to access, download, copy, distribute, and use these materials in any way, even commercially, with proper attribution. For these reasons, we cannot publish previously copyrighted maps or satellite images created using proprietary data, such as Google software (Google Maps, Street View, and Earth). For more information, see our copyright guidelines: http://journals.plos.org/plosone/s/licenses-and-copyright.

Reviewers' comments:

Reviewer's Responses to Questions

**Comments to the Author**

1. Is the manuscript technically sound, and do the data support the conclusions?

Reviewer #1: Yes

2. Has the statistical analysis been performed appropriately and rigorously? 

Reviewer #1: Yes

3. Have the authors made all data underlying the findings in their manuscript fully available?

Reviewer #1: No

4. Is the manuscript presented in an intelligible fashion and written in standard English?

Reviewer #1: Yes

5. Review Comments to the Author

Reviewer #1: In this paper, the authors conducted a qualitative study on health professionals' perspectives towards the role of mHealth in improving maternal and child health in Southern Ethiopia. The paper is well-written and easy to follow, addressing an interesting subject with great potential for facilitating the adoption of mHealth in developing countries like Ethiopia. However, there are some areas that need improvement and clarification:

1- Define what mHealth is. Readers might have no idea what you are talking about.

2- Introduce the complete word for mHealth (Mobile health) for the first time that you used this abbreviation.

3- The paper misses a good and comprehensive related work section. Currently, related work is limited and mixed with introduction. I think it would be better if you separate the related work from the introduction, create a section specific to the related work, and completely explain the applications and benefits that mHealth solutions have had in this subject.

4- Did you conduct the interviews and FGD in-person or online? Please explain this in the paper.

5- I did not understand what this number is in front of the quotes e.g. (II 4.1) or (KII 7.2). I think it would be better if you write (KII, his/her profession, sex, age)

5- What is a mother forum? Is it some kind of in-person meetings that are held for the mothers? Please explain that in the paper.

6- The discussion needs to be revised. Now, it is more like the results section, like you are reporting the results again. You need to interpret your results with a broad perspective with the help of others’ work in the discussion section.

Please read this instruction for rewriting the discussion.

"The discussion reviews the findings and puts them into the context of the overall research. It brings together all the sections that came before it and allows a reader to see the connections

between each part of the research paper. In a discussion section, the author engages in three necessary steps: interpretation, analysis, and explanation. An effective discussion section will tell

a reader why the research results are important and where they fit in the current literature, while also being self-critical and candid about the shortcomings of the study."

6. PLOS authors have the option to publish the peer review history of their article (what does this mean?). If published, this will include your full peer review and any attached files.

Reviewer #1: No

---

## [Author Response · Author response to Decision Letter 0]

7 Oct 2023

Editorial comments

Comments: We note that the grant information you provided in the ‘Funding Information’ and ‘Financial Disclosure’ sections do not match.

Response: Thank you for the comment. we insert the grant number in Financial Disclosure section.

Comments: Please include your full ethics statement in the ‘Methods’ section of your manuscript file. In your statement, please include the full name of the IRB or ethics committee who approved or waived your study, as well as whether or not you obtained informed written or verbal consent. If consent was waived for your study, please include this information in your statement as well.

Response: Thank you for the comment. We improved ethical approval (page 7, line195-196)

Comments: Please ensure that you refer to Figure 1 in your text as, if accepted, production will need this reference to link the reader to the figure.

Responses: Thank you for the comment. There is no figure in the document anymore.

Reviewers’ comments

Reviewer#1

Comment: Define what mHealth is. Readers might have no idea what you are talking about.

Response: Thank you for the comments. We defined mHealth (page 3, line 81-84)

Comment: Introduce the complete word for mHealth (Mobile health) for the first time that you used this abbreviation.

Responses: Thank you for the comments. We defined mHealth (page 3, line 81)

Comment: The paper misses a good and comprehensive related work section. Currently, related work is limited and mixed with introduction. I think it would be better if you separate the related work from the introduction, create a section specific to the related work, and completely explain the applications and benefits that mHealth solutions have had in this subject.

Response: Thank you for the comments. We completely modified the introduction section short and precise (page 3-4, line 69-107)

Comment: Did you conduct the interviews and FGD in-person or online? Please explain this in the paper.

Response: Thank you for the comments. Yes, it is presented under Source Population and sample size section (page 5, line 141-151)

Comment: I did not understand what this number is in front of the quotes e.g. (II 4.1) or (KII 7.2). I think it would be better if you write (KII, his/her profession, sex, age)

Response: Thank you for the comments. Of course the numbers are the reference to the source code in the original source. They are traceable. We appreciate your comments and changed this format (page 10-17 line 237-495)

Comment: What is a mother forum? Is it some kind of in-person meetings that are held for the mothers? Please explain that in the paper.

Responses: Thank you for the comments. We added mothers forum definition(page 12, line 296-298)

comments: The discussion needs to be revised. Now, it is more like the results section, like you are reporting the results again. You need to interpret your results with a broad perspective with the help of others’ work in the discussion section. 

Please read this instruction for rewriting the discussion.

"The discussion reviews the findings and puts them into the context of the overall research. It brings together all the sections that came before it and allows a reader to see the connections

between each part of the research paper. In a discussion section, the author engages in three necessary steps: interpretation, analysis, and explanation. An effective discussion section will tell

a reader why the research results are important and where they fit in the current literature, while also being self-critical and candid about the shortcomings of the study."

Responses: Thank you for the comments. The discussion is completely changed and more sound now.

---

## [Editor Report · Decision Letter 1]

16 Nov 2023

PONE-D-23-11693R1The Role Of mHealth Intervention to Improve Maternal and Child Health: A Provider-Based Qualitative Study in Southern Ethiopia.PLOS ONE

Dear Dr. Girma Gilano

Thank you for submitting your manuscript to PLOS ONE. After careful consideration, we feel that it has merit but does not fully meet PLOS ONE’s publication criteria as it currently stands. Therefore, we invite you to submit a revised version of the manuscript that addresses the points raised during the review process.

We look forward to receiving your revised manuscript.

Kind regards,

Daniel Dramani Kipo-Sunyehzi, PhD, MPhil, MA, BA

Academic Editor

PLOS ONE

Reviewers' comments:

Reviewer's Responses to Questions

Comments to the Author

1. Is the manuscript technically sound, and do the data support the conclusions?

Reviewer #1: Yes

2. Has the statistical analysis been performed appropriately and rigorously?

Reviewer #1: Yes

3. Have the authors made all data underlying the findings in their manuscript fully available?

Reviewer #1: No

4. Is the manuscript presented in an intelligible fashion and written in standard English?

Reviewer #1: Yes

5. Review Comments to the Author

Reviewer #1: In this paper, the authors conducted a qualitative study on health professionals' perspectives towards the role of mHealth in improving maternal and child health in Southern Ethiopia. The paper is well-written and easy to follow, addressing an interesting subject with great potential for facilitating the adoption of mHealth in developing countries like Ethiopia. However, there are some areas that need improvement and clarification:

1- Define what mHealth is. Readers might have no idea what you are talking about.

2- Introduce the complete word for mHealth (Mobile health) for the first time that you used this abbreviation.

3- The paper misses a good and comprehensive related work section. Currently, related work is limited and mixed with introduction. I think it would be better if you separate the related work from the introduction, create a section specific to the related work, and completely explain the applications and benefits that mHealth solutions have had in this subject.

4- Did you conduct the interviews and FGD in-person or online? Please explain this in the paper.

5- I did not understand what this number is in front of the quotes e.g. (II 4.1) or (KII 7.2). I think it would be better if you write (KII, his/her profession, sex, age)

5- What is a mother forum? Is it some kind of in-person meetings that are held for the mothers? Please explain that in the paper.

6- The discussion needs to be revised. Now, it is more like the results section, like you are reporting the results again. You need to interpret your results with a broad perspective with the help of others’ work in the discussion section.

Please read this instruction for rewriting the discussion.

"The discussion reviews the findings and puts them into the context of the overall research. It brings together all the sections that came before it and allows a reader to see the connections

between each part of the research paper. In a discussion section, the author engages in three necessary steps: interpretation, analysis, and explanation. An effective discussion section will tell

a reader why the research results are important and where they fit in the current literature, while also being self-critical and candid about the shortcomings of the study."

6. PLOS authors have the option to publish the peer review history of their article (what does this mean?). If published, this will include your full peer review and any attached files.

Do you want your identity to be public for this peer review? For information about this choice, including consent withdrawal, please see our Privacy Policy.

Reviewer #1: No

---

## [Author Response · Author response to Decision Letter 1]

17 Nov 2023

Editorial comments

Comments: We note that the grant information you provided in the ‘Funding Information’ and ‘Financial Disclosure’ sections do not match.

Response: Thank you for the comment. we insert the grant number in Financial Disclosure section.

Comments: Please include your full ethics statement in the ‘Methods’ section of your manuscript file. In your statement, please include the full name of the IRB or ethics committee who approved or waived your study, as well as whether or not you obtained informed written or verbal consent. If consent was waived for your study, please include this information in your statement as well.

Response: Thank you for the comment. We improved ethical approval (page 7, line195-196)

Comments: Please ensure that you refer to Figure 1 in your text as, if accepted, production will need this reference to link the reader to the figure.

Responses: Thank you for the comment. There is no figure in the document anymore.

Reviewers’ comments

Reviewer#1

Comment: Define what mHealth is. Readers might have no idea what you are talking about.

Response: Thank you for the comments. We defined mHealth (page 3, line 81-84)

Comment: Introduce the complete word for mHealth (Mobile health) for the first time that you used this abbreviation.

Responses: Thank you for the comments. We defined mHealth (page 3, line 81)

Comment: The paper misses a good and comprehensive related work section. Currently, related work is limited and mixed with introduction. I think it would be better if you separate the related work from the introduction, create a section specific to the related work, and completely explain the applications and benefits that mHealth solutions have had in this subject.

Response: Thank you for the comments. We completely modified the introduction section short and precise (page 3-4, line 69-107)

Comment: Did you conduct the interviews and FGD in-person or online? Please explain this in the paper.

Response: Thank you for the comments. Yes, it is presented under Source Population and sample size section (page 5, line 141-151)

Comment: I did not understand what this number is in front of the quotes e.g. (II 4.1) or (KII 7.2). I think it would be better if you write (KII, his/her profession, sex, age)

Response: Thank you for the comments. Of course the numbers are the reference to the source code in the original source. They are traceable. We appreciate your comments and changed this format (page 10-17 line 237-495)

Comment: What is a mother forum? Is it some kind of in-person meetings that are held for the mothers? Please explain that in the paper.

Responses: Thank you for the comments. We added mothers forum definition(page 12, line 296-298)

comments: The discussion needs to be revised. Now, it is more like the results section, like you are reporting the results again. You need to interpret your results with a broad perspective with the help of others’ work in the discussion section. 

Please read this instruction for rewriting the discussion.

"The discussion reviews the findings and puts them into the context of the overall research. It brings together all the sections that came before it and allows a reader to see the connections

between each part of the research paper. In a discussion section, the author engages in three necessary steps: interpretation, analysis, and explanation. An effective discussion section will tell

a reader why the research results are important and where they fit in the current literature, while also being self-critical and candid about the shortcomings of the study."

Responses: Thank you for the comments. The discussion is completely changed and more sound now.

---

## [Editor Report · Decision Letter 2]

27 Nov 2023

The Role Of mHealth Intervention to Improve Maternal and Child Health: A Provider-Based Qualitative Study in Southern Ethiopia.

PONE-D-23-11693R2

Dear Dr. Girma Gilano 

We’re pleased to inform you that your manuscript has been judged scientifically suitable for publication and will be formally accepted for publication once it meets all outstanding technical requirements.

Kind regards,

Daniel Dramani Kipo-Sunyehzi, PhD, MPhil, MA, BA

Academic Editor

PLOS ONE

Additional Editor Comments (optional):

Good responses
---

## [Editor Report · Acceptance letter]

30 Jan 2024

PONE-D-23-11693R2 

PLOS ONE

Dear Dr. Gilano, 

I'm pleased to inform you that your manuscript has been deemed suitable for publication in PLOS ONE. Congratulations! Your manuscript is now being handed over to our production team.

Kind regards, 

on behalf of

Dr Daniel Dramani Kipo-Sunyehzi 

Academic Editor

PLOS ONE